# Prediction of Total Soluble Solids and pH of Strawberry Fruits Using RGB, HSV and HSL Colour Spaces and Machine Learning Models

**DOI:** 10.3390/foods11142086

**Published:** 2022-07-13

**Authors:** Jayanta Kumar Basak, Bolappa Gamage Kaushalya Madhavi, Bhola Paudel, Na Eun Kim, Hyeon Tae Kim

**Affiliations:** 1Institute of Smart Farm, Gyeongsang National University, Jinju 52828, Korea; jk.basak@gnu.ac.kr; 2Department of Environmental Science and Disaster Management, Noakhali Science and Technology University, Noakhali 3814, Bangladesh; 3Department of Bio-Systems Engineering, Institute of Smart Farm, Gyeongsang National University, Jinju 52828, Korea; kaushalyamadhavi81@gmail.com (B.G.K.M.); bholapaudel@gnu.ac.kr (B.P.); na.eun@gnu.ac.kr (N.E.K.)

**Keywords:** colour spaces, image processing technique, multiple linear regression, pH, strawberry, support vector machine regression, total soluble solids

## Abstract

Determination of internal qualities such as total soluble solids (TSS) and pH is a paramount concern in strawberry cultivation. Therefore, the main objective of the current study was to develop a non-destructive approach with machine learning algorithms for predicting TSS and pH of strawberries. Six hundred samples (100 samples in each ripening stage) in six ripening stages were collected randomly for measuring the biometrical characteristics, i.e., length, diameters, weight and TSS and pH values. An image of each strawberry fruit was captured for colour feature extraction using an image processing technique. Channels of each colour space (RGB, HSV and HSL) were used as input variables for developing multiple linear regression (MLR) and support vector machine regression (SVM-R) models. The result of the study indicated that SVM-R model with HSV colour space performed slightly better than MLR model for TSS and pH prediction. The HSV based SVM-R model could explain a maximum of 84.1% and 79.2% for TSS and 78.8% and 72.6% for pH of the variations in measured and predicted data in training and testing stages, respectively. Further experiments need to be conducted with different strawberry cultivars for the prediction of more internal qualities along with the improvement of model performance.

## 1. Introduction

Strawberry (*Fragaria x ananassa*) is widely consumed because of its distinctive flavour and appearance, and it was found to have the potential ability to prohibit the growth of cancer cells [1,2,3]. Moreover, the decision for subsequent consumption of strawberries also depends on the internal quality of fruits, which is related to the total soluble solids (TSS) content, acidity (pH) and one of the most important vitamin C [4]. With the improvement in the quality of strawberry fruits, the determination of internal quality parameters, such as TSS and pH, is a paramount concern. Therefore, developing an effective and simple method for assessing the interior properties of strawberries is essential to provide consumers with consistent, safe and nutritious fruit.

TSS and pH are the important internal parameters of strawberries for determining their maturity and taste [5]. The levels of these two chemical properties also influence the crop values and the acceptance level by consumers [6]. TSS in strawberries mainly contains sugars and acids with small quantities of dissolved vitamins, fructans, proteins, pigments, phenolics and minerals [7,8]. Sugar content in strawberries is implicated in flavour and determine caloric value of fruits [9]. pH in strawberries, which mainly comes from citric acid, plays a significant role in escalating the ripening of the fruit [10]. In addition, ascorbic acid is one of the most important organic acids involved in flavour, texture, pH and colour of the strawberry [9]. The TSS and pH attributes are significantly affected by the different levels of ripening stages [6]. As a non-climacteric fruit, the full ripening stage is usually chosen for its harvesting since it does not gain desirable properties suited for fresh consumption before detachment [11]. With the changing of ripening stages from white ripening to dark red, significant variations of TSS and pH levels with the fruit’s colour were observed in several strawberry cultivars [12,13]. Likewise, large variations of the fruit’s colour, TSS and pH, according to the different ripening stages of strawberries, were also reported by Martínez-Bolaños et al. [14]. Therefore, the determination and quantification of TSS and pH with considering the fruit’s colour in different ripening stages are of key concern in food sciences research. 

The TSS and pH of strawberries are commonly measured by instrumental assessment along with sensory evaluation methods [4,15]. Hydrometer, refractometer and high-performance chromatography are available instruments widely used for assessing the TSS level, while pH meter based on the potentiometric/electrometric techniques are commonly accepted and used for determining pH levels [16,17,18,19,20]. A hydrometer is a simple and common instrument for quantifying TSS. However, it is inadequate and inaccurate due to the incorrect reading associated with huge variations in surface tension on account of surface contamination [21]. Different types of refractometers have also been used for measuring the internal qualities of strawberry fruits, which are more suitable and reliable because they are not affected by the suspended solids, nevertheless, sometimes errors may happen if the readings are taken rapidly [17]. High-performance chromatography techniques are the most effective and innovative method in measuring high accuracy results; however, it is worth noting that the method is very complex and not suitable for quantifying internal quality parameters in a wide range in field levels [21]. Apart from instrumental techniques, sensory evaluation methods, such as triangle test, duo-trio test and paired comparison test, are traditionally used to evaluate the subjective based result other than quantitative and do not give insight into the wider applicability [22]. In addition, all instrumental assessment techniques and sensory evaluation methods are time consuming procedures and those are more applicable in laboratory experiments compared to the real fields. 

Recently, the improvement of non-destructive technologies, such as Visible/near-infrared (Vis/NIR) spectroscopy, computer vision and hyperspectral imaging, have gained more attention for measuring the internal qualities of fruits. Among the non-destructive technologies, Vis/NIR spectroscopy is one of the promising analytical methods where pre-sampling is not essential for determining the fruit’s quality [23]. Several studies have been conducted using Vis/NIR spectroscopy to assess the chemical properties of fruits including apples, kiwifruits, tomatoes, oranges, mangoes, mandarins, plums, watermelons and apricot [23,24,25,26,27,28]. Recently, the developed version of Vis/NIR techniques is also being used for sorting different types of fruits based on their internal quality parameters [29]. However, one of the major problems of using this method is the thickness of fruit rinds; less accuracy is obtained for high skin thickness of fruits like watermelon and citrus [21]. Apart from the Vis/NIR spectroscopy technique, several imaging techniques such as multi- and hyper-spectral, thermal imaging techniques have been developed and applied successfully for the quality assessment of a variety of fruits [30]. Nowadays, fruits can be easily inspected and assessed internal parameters using multi- and hyper-spectral imaging techniques. Over the past decade, several studies were reported for quality assessment including dry matter, TSS and pH of strawberries using multi- and hyper-spectral imaging techniques [31,32]. In addition, laser-induced method, thermal imaging, photoacoustic spectroscopy or imaging, X-ray techniques and odour imaging non-destructive techniques have been used for the qualitative detection of fruits. However, the main limitations of these non-destructive technologies are highly expensive equipment needs to be used and the procedures are very complex which are sometime inconceivable for farmhands for fruits’ quality detection and assessment. Therefore, in this study, we have introduced a simple and inexpensive method where we applied three colour spaces, i.e., Red, Blue and Green (RGB); Hue, Saturation and Value (HSV); and Hue, Saturation and Lightness (HSL), acquired by image processing technique and developed prediction models using machine learning algorithms for measuring TSS and pH of strawberries. 

According to the previous literature, RGB, HSV and HSL colour spaces have been extensively used for sorting strawberries [33,34,35]. Pardede et al. [34] used four colour spaces, i.e., RGB, HSV, HSL and L * a * b *, and support vector machine (SVM) algorithms to evaluate the level of ripeness of fruits, and based on the experimental results, it was shown that HSV colour feature achieved the best accuracy levels in determining the ripening stages of fruits. In another study, SVM and Caffe-Net algorithms with fast recognition methods were used to evaluate the maturity level of strawberries [36]. Hariri et al. [37] investigated the ripeness level of bell pepper by using a content-based image classification system in different maturity stages and obtained a classification accuracy of 93.89%. A multi-class classification approach and SVM with linear kernel were applied for tomato ripeness measurement where the ripeness classification accuracy had 90.80% [38]. From the available literature [39,40,41], maximum experiments were conducted on various image classification techniques and SVM algorithms to determine the ripeness level of fruits. Hence, the present study was conducted to establish relationships and to develop models between colour spaces, i.e., RGB, HSV and HSL, and fruit’s quality parameters (TSS and pH) using a linear model, i.e., multiple linear regression (MLR) and a non-linear model, i.e., support vector machine regression (SVM-R). This non-destructive method minimizes sampling efforts and expenses as well as improves precision where the samples are complicated to handle. Hence, the main objectives of this study are (1) to determine the TSS and pH level in the six ripening stages of strawberry fruits; (2) to acquire RGB, HSV and HSL colour space values using an image processing technique; (3) to develop MLR and SVM-R models using the colour space’s values for predicting TSS and pH values of strawberries and, finally, (4) to analyse the sensitivity of the best colour space’s values in determining influential colour channels on TSS and pH prediction.

## 2. Materials and Methods

### 2.1. Plant Materials and Sample Collection

The Seolhyang strawberry (*Fragaria x ananassa Duch.*) plants were grown in a greenhouse at Smart Farm Systems Laboratory at Gyeongsang National University in Korea from September 2021 to January 2022. The air temperature inside the greenhouse was controlled with a Farmlink controller system (Farmlink™ v 3.0, Jinju, Gyeongsangnam-Do, South Korea) during all growing stages, allowing temperatures only to range from 10 to 30 °C [6]. Moreover, other environmental parameters inside the greenhouse such as CO_2_, humidity and light were monitored using a highly accurate sensor unit MCH 383SD (Lutron Electronic Enterprises Co. Ltd., Taipei, Taiwan). A total of five hundred strawberry plants in five rows have been cultivated under the combination of bio plus compost soil and Hoagland solution (Figure 1). Bio plus compost soil is very famous for greenhouse crop cultivation in Korea which consist of cocopeat (68.86%), peat moss (11.00%), perlite (11.00%) and zeolite (9.00%) [42]. Irrigation was applied daily in the range from 20 to 30 mL per plant in early stage to 30 to 50 mL per plant in ripening stage of strawberries [43]. 

Strawberries were harvested at six ripening stages, i.e., Whiting (RS1), Turning (RS2), Half-red (RS3), Three-quarter red (RS4), Bright red (RS5) and Dark red (RS6) based on the colour development on skin, ranging from white to dark red [6]. The typical samples of strawberries of the six ripening stages as shown in Figure 2. In each ripening stage, 100 fruits were collected and numbered for biometrical characteristics, TSS and pH determination and image acquisition.

### 2.2. Determination of Biometrical Characteristics, Total Soluble Solid and pH 

Before the determination of biometrical characteristics, TSS and pH, each fruit was washed with water, drained and dried with paper towels first. Biometrical characteristics including weight, diameters and length were measured on 100 fruits at each ripening stage. For measuring the fresh weight (in g) of fruits, a digital balance (FX-300iWP, A&D Company Ltd., Tokyo, Japan) was used. A digital Vernier caliper (Digital Caliper E03-150 122-521, Datac Co., Ltd., Seoul, Korea) was used to determine the stem length and diameters (in mm). Table 1 shows the summary statistics of fruits’ weight, length and diameters in six ripening stages.

TSS and pH were measured directly using a portable digital refractometer (Model: KERN ORA 32BA, KERN & SOHN GmbH, Balingen, Germany) and a digital pH meter (Model: HM Digital pH-200 Waterproof Temperature pH Meter, HM Digital, Daejon, Korea), respectively. Extracted juice prepared using a mechanical blender (Model: TT-I777, Two thousand Machinery Co., Ltd., Guangzhou, China) from individual fruit was used to determine the pH according to the method of AOCC [44], while the TSS was measured using the procedure described by Dadzie and Orchard [45]. The summary statistics of TSS and pH of fruits in each ripening stage are presented in Table 1.

### 2.3. Image Acquisition of Strawberry Fruits

A laboratory-based imaging system was applied to acquire strawberries images. To acquire the images, each strawberry sample was placed in a smooth rectangular light chamber (80 × 80 × 80 cm). The light chamber consisted of two strips of light-emitting diodes (LEDs) and the total light emitting capacity of lamps in each strip was 10 W. The inner part of the light chamber (except the bottom surface) consisted of an aluminium surface, reflecting the light from all directions to ensure no shadow forms around the strawberry and got high-quality images. The bottom surface inside the light chamber consists of a black surface for an even background. An RGB camera (SONY DSC-RX100 vii, Sony, Seoul, Korea) with 5472 × 3648 pixels resolution was used for capturing images. The distance between the camera lens and the sample plate was fixed as 80 cm. Images were captured on the two opposite sides in horizontal plane of each one of the 600 strawberries in the six ripening stages (Figure 2). During the image acquisition time, the procedures were kept the same for all strawberries.

### 2.4. Colour Feature Extraction of Strawberry Fruits

The acquired images were processed using open-source libraries in a Python (Python 3.7.0) environment. Our region of interest (ROI) was only the fruit parts, therefore, each image sample of strawberries was segmented from its background using remove.bg software and resized it by 500 × 500 pixels. After background removal from each image, red (*R*), green (*G*) and blue (*B*) values were computed using a channel splitting algorithm developed using Python. RGB colour space is a common colour standard for obtaining a wide array of colours where each channel ranges from 0 to 255 [46]. Subsequently, each RGB image was converted to HSV (Hue, Saturation and Value) and HSL (Hue, Saturation and Lightness) colour spaces. The HSV colour space represents the nuances of colour in 3-D cylindrical coordinates which is often called the hexcone model. The HSL colour space was developed with the same idea as the HSV colour space with 3-D cylindrical coordinates; however, due to consisting of 2 cones, HSL model is called the bi-hexconemodel [34]. The following equations have been used to transfer of RGB colour space into the HSV (Equations (1)–(3)) and HSL (Equations (4) and (6)) colour spaces [47,48].
(1)V=Max (R, G, B)
(2)$$H=\{\begin{array}{c} undefined,\;if\;Max=Min\\ \left(\frac{G-B}{Max-Min}\right)\times A,\;if\;Max=R\\ \left(\frac{B-R}{Max-Min}+2\right)\times A,\;if\;Max=G\\ \left(\frac{R-G}{Max-Min}+4\right)\times A,\;if\;Max=B \end{array}$$
(3)$$S=\{\begin{array}{c} 0,\;if\;Max=Min\\ \left(Max-Min\right),\;others \end{array}$$
where in RGB colour space, *R*, *G* and *B* values vary from 0 to 255. *Max* = max (*R*, *G*, *B*), *Min* = min (*R*, *G*, *B*) and *A* = π/3 if *H* is expressed in radians or *A* = 60° if *H* is expressed in degrees. The range of *H* varies from −π/3 to +5π/3 in radians scale and −60° to 300° in degree scale. Whereas the *S* and *V* vary from 0 to 255.
(4)$$L=\frac{Max+Min}{2}$$
(5)$$H=\{\begin{array}{c} undefined,\;if\;Max=Min\\ \left(\frac{G-B}{Max-Min}\right)\times 60{}^{\circ},\;if\;Max=R\\ \left(\frac{B-R}{Max-Min}+2\right)\times 60{}^{\circ},\;if\;Max=G\\ \left(\frac{R-G}{Max-Min}+4\right)\times 60{}^{\circ},\;if\;Max=B \end{array}$$
(6)S={0, if Max=MinMax−Min2L255, if L≥127 Max−Min2−(2L255), if L<127
where *Max* = max (*R*, *G*, *B*) and *Min* = min (*R*, *G*, *B*). The range of *H* varies from −π/3 to +5π/3 in radians scale, while the *S* and *L* values vary from 0 to 255.

### 2.5. Data Pre-Processing for Prediction Model Development

As a first step, overall datasets, i.e., *R*, *G*, *B* values from RGB colour space; *H*, *S*, *V* values from HSV colour space; and *H*, *S*, *L* values from HSL colour space, and TSS and pH were obtained in six ripening stages of 600 strawberries. In the next step, the measured data were subjected to the data preprocessing techniques, including missing data analysis, feature extraction and data normalization, as well as training and testing data partition. In the measured data, there were no missing values; therefore, the current research did not consider any method for imputing the missing data. A rank correlation test was widely used to identify the right features from the available datasets [49]. In this study, rank correlation, i.e., the Pearson correlation technique, was applied, as briefly described in the input datasets for model development section. A different range of variables in a same dataset used as an input parameters for developing machine learning (ML) models may reduce the model’s learning performance and prediction efficiency [49]. For that reason, data normalization, a popular data preprocessing technique, has been commonly used to avoid complications. In this study, the *Z*-score data normalization technique (Equation (7)) has been applied to keep data in the range from 0 to 1 within the same scale that was used across all numeric columns in datasets.
(7)$$Z=\frac{x-\mu }{\sigma }$$
where Z, x, μ and σ denote the standard score, value in the data set, mean of all values in the data set and standard deviation, respectively.

Moreover, the prediction capabilities of ML models are also influenced by the data partition in the training and testing stages. Several studies used 70:30 (training: testing), 80:20 and 90:10 partition to develop and validate model [49,50,51,52]. After examining the model performance using the three data partitions (70:30, 80:20 and 90:10), the current study utilized 80% data during the training stage and 20% data during the testing stage.

### 2.6. Development of Multiple Linear Regression (MLR)

MLR model is developed using the same principle as simple linear regression where the analysis is carried out to predict one dependent variable for a given set of independent variables [53]. MLR offers a wide range of real-world applications in three main areas: evaluating relationships between variables, numerical predictions and time series forecasting. It is widely used in various fields including yield prediction of crops, weather forecasting, demand calculation of electricity, business forecast, etc. [52,54,55]. The formula of MLR model is represented by the Equation (8).
(8)$$Y_{i}=\beta_{o}+\beta_{1}X_{1}+\beta_{2}X_{2}+\ldots \ldots \ldots .\;..+\beta_{n}X_{n}+\varepsilon_{i}$$
where $Y_{i}$ is the predicted variable (TSS/pH) in this formula, *β*_0_−*β_n_* are the coefficients of regression, *X*_1_−*X_n_* are the input variables (*R*, *G*, *B* values for RGB colour space; *H*, *S*, *V* values for HSV colour space and *H*, *S*, *L* values for HSL colour space), and *ε* is the error associated with the *i*th observation. The basic architecture of MLR is shown in Figure 3a.

### 2.7. Development of Support Vector Machine Regression (SVM-R)

SVM is a powerful non-linear multivariate technique that creates a hyperplane or set of hyperplanes in a high-dimensional space for classification named support vector classification (SVC), and the same principle is applied for support vector machine regression (SVM-R) [56,57]. Using kernel functions in non-linear SVM is important since SVM effectiveness is influenced by the kernel function employed in the model [47]. The SVM model is built using three basic kernel functions: polynomial, radial basis and sigmoid functions, with the radial basis function (RBF) being the most commonly employed due to its ability to handle non-linear relationships more efficiently compared to others [23]. After experimenting on several SVM-R structures using the three kernel functions with gamma (γ), epsilon (ε) and penalty parameter of the error term (C), the study decided to apply radial basis function (RBF) as kernel type, γ = 0.5, ε = 0.1 and C = 50. In this study, the predicted TSS and pH values of strawberries were followed by the SVM-R operated formula (Equation (9)).
(9)$${\hat{Y}}_{0}=\overset{n}{\underset{i=1}{\sum }}K\left(X_{i},X_{0}\right)\left(\alpha_{i}-\alpha_{i}^{*}\right);K\left(X_{i},X_{0}\right)=\frac{\exp{\left(-\left|X_{i}-X_{j}\right|\right)}^{2}}{Y}$$
where $\alpha_{i}$ and $\alpha_{i}^{*}$ denote the support vectors and $K\left(X_{i},X_{0}\right)$ represents the radial basis function. The basic architecture of the one-dimensional SVM-R is shown in Figure 3b.

### 2.8. Application Methodology and Performance Metrics

In this study, MLR and SVM-R models were developed employing open-source libraries in Python (Python 3.7.0) environment. As a high-level programming language, python was extensively used in scientific fields. Data processing, manipulation and visualization were accompanied using NumPy [58], Pandas [59] and Matplotlib [60], respectively in the python environment. Root mean square error (RMSE) (Equation (10)) and coefficient of determination (R^2^) (Equation (11)) evaluation metrics have been used to measure the performance of the two models. Statistical Package for the Social Sciences (IBM, SPSS Statistics 22.0.0.0, New York, NY, USA) was used for statistical analysis and Origin Pro 9.5.5 (OriginLab, Northampton, MA, USA) was used for the data representation as figures. Figure 4 shows the aggregate workflow of the current study systematically.
(10)$$RMSE=\sqrt{\frac{1}{n}\Sigma_{t=1}^{n}\;{\left(y_{t\;actual-}y_{t\;predicted}\right)}^{2}}$$
(11)$$R^{2}=1-\frac{\Sigma_{t=1}^{n}{\left(y_{t\;actual-}y_{t\;predicted}\right)}^{2}}{\Sigma_{t=1}^{n}{\left(y_{t\;actual-}y_{t\;mean}\right)}^{2}}$$

## 3. Results and Discussion

### 3.1. Changes in Biometrical Characteristics, Total Soluble Solid and pH 

In this study, a total of 600 samples were collected for analysing biometrical characteristics (length, diameter and weight), total soluble solids (TSS) and pH in six different ripening stages of strawberries. Table 1 shows the summary of statistics of each measurement for all datasets. Length, diameter, weight, TSS and pH of strawberries changed significantly in different repining stages and the variation was particularly observed in all stages for TSS and pH measurement. In addition, the length, diameter and weight of strawberries did not change significantly in all ripening stages at the same levels. Analysing the 100 samples in each ripening stage, significant variations were observed among white (RS1), half red (RS3) and bright red (RS5) stages for length, diameter and weight of strawberries. These observations were expected since growing and ripening occur simultaneously in strawberry fruits [61]. Abeles and Takeda [62] reported an increase in growing phase of ‘Tribute’ strawberries from white to red ripening stage. Strawberry varieties such as ‘Pajaro’, ‘Chandler’ and ‘Selva’ were shown the similar patterns [63,64,65]. Length, diameter and weight of strawberry fruit are scarcely mentioned in this current study. In general, strawberries collected in different ripening stages showed good biometrical characteristics.

TSS in strawberries increased continuously from the white to dark red ripening stage with statistically significant. TSS in tested strawberries (TSS = 4.90–10.15 °brix) was into the range (TSS = 4.8–10.9 °brix) reported in literature for the ripening stages of strawberries [12,66]. Similar results were mentioned for different strawberry cultivars in the ripening stages [13,67]. During the ripening stages, the fruit’s starch degrades and transforms into TSS components by the hydrolysis of enzymes, i.e., α-amylase and β-amylase [68]. Villanueva et al. [69] stated that total sugar levels rise as the strawberries ripen, whereas invertase enzyme activity declines. Like TSS, pH levels also increased significantly in the ripening stages, and their changes were highly correlated with TSS (r = 0.76). The content of pH in tested fruits (pH = 3.45–5.32) was into the range (pH = 3.20–5.40) reported for fruit from many strawberry cultivars, including ‘Daewang’, ‘Maehyang’ and ‘Santa’ [70]. Montero et al. [9] reported that the pH levels increased continuously for ‘Chadler’ strawberries in the ripening period. Similar results were mentioned for ‘Oso Grande’ and ‘Sweet Charly’ strawberries [11]. The increasing trend of a strawberry’s pH level may occur due to the breaking down of organic acids and converting into sugar after the respiration process [71]. As a result, the TSS and pH values increased with the decreased of organic acids such as malic or citric acid, where these acids are the primary substrates for the respiration process in ripening period [72]. 

### 3.2. Changes in Channels of RGB, HSV and HSL Colour Spaces

Based on the analysis of variance (ANOVA), the study results found that strawberries at six ripening stages had high significant variations of colour values (Table 2, Table 3 and Table 4). The colour of strawberries develops from white to dark red during its ripening process [73]. In RGB colour space, *R* values increased significantly in RS2, RS3, RS4 and RS5 stages compared to RS1, however, in the RS6 stage, remarkably, it reduced. In addition, *G* and *B* values gradually decreased from the white to dark red stage and the variations were found significantly in all ripening stages of strawberries (Table 2). Similar changes in colour spaces were reported for ripening ‘Selva’ strawberries [63]. In the case of HSV colour space, Hue (H) colour channel in strawberries diminished continuously from RS1 to RS5. Initially, *H* value was calculated 65% lower at the RS5 stage compared to RS1.

Ferreyra et al. [63] and de Jesús Ornelas-Paz et al. [6] also reported a continuous decrease of H in ripening stages. In addition, *S* values significantly reduced in all ripening stages, whereas *V* values showed a different level of change (Table 3). Like HSV colour space, a similar pattern of changes has been observed in HSL colour space (Table 4). In general, *S* and *L* values of sampled ripe fruit (RS6; *S* = 109.21 ± 4.16 and *L* = 195.71 ± 7.24) were in the typical range (*S* = ~100.00–220.00 and *L* = ~120.00–230.00) reported in literatures during different ripening stages of strawberries [11,74]. Similar findings were stated in ripening fruits from other strawberry cultivars [12].

### 3.3. Input Datasets for Model Development 

The selection of input variables is one of the most important aspects of developing machine learning models as well as improving model accuracy [50]. There are various approaches to find out the most suitable variables; however, the correlation coefficient is one of the most effective methods for this subject [75]. In the current study, we used the Pearson correlation coefficient technique, which explained the explanatory and response variables and determined the strength of a relationship between those variables. Having a good relationship, the independent variables can be considered a strong predictor of the dependent variables [76]. The relationship between the independent variables, i.e., RGB (red, green and blue), HSV (Hue, Saturation and Value) and HSL (Hue, Saturation and Lightness) colour spaces and dependent variables (TSS and pH), are illustrated using heat maps (Figure 5).

As shown in Figure 5, there was a negative correlation between RGB colour space and TSS and pH values. The findings of the study also revealed that G was more influential factor compared to R and B channels in predicting TSS (r = −0.81) and pH (r = −0.75) values. Moreover, a high positive correlation was found between the S channel and TSS (r = 0.90) and pH (r = 0.72) values in HSV colour space, whereas a negative correlation was exhibited between H channel and TSS and pH values. Like H channel, V channel also showed a negative relationship with TSS and pH values. According to the heat map results (Figure 5), in HSL colour space, H and S channels showed a strong negative relationship with TSS and pH, however, L had a positive correlation with the two factors. Based on the overall analysis, it can be concluded that S in HSV colour space had the highest correlation value with TSS and pH among all input channels. In addition, all channels in each colour space had a good correlation (r ≥ 0.5) with TSS and pH; therefore, every channel was considered as an input variable in developing machine learning models in the present study.

### 3.4. MLR Model Performance for TSS and pH Prediction 

The existence of linear relationships between dependent and independent variables is essential for developing regression-based models to predict outcomes [50]. A number of studies used simple linear models due to their simplicity of nature and easily interpretation of outcomes [42,52,76]. In the present study, we used a linear regression model, i.e., multiple linear regression (MLR) model, in predicting TSS and pH values. Channels of the three colour spaces, i.e., RGB, HSV and HSL, were used as input variables for developing the MLR model. The performances of MLR model to predict TSS and pH are shown in Table 5 and Table 6.

In terms of TSS prediction, the highest R^2^ was observed with 0.821 (training phase) and 0.763 (testing phase) for HSV colour space, indicating that the MLR model can explain a maximum of 82.1% and 76.3% of the variations in the measured and predicted data in training and testing period, respectively. However, the lowest R^2^ and the highest RMSE were found in training (R^2^ = 0.768 and RMSE = 0.532 °brix) and testing (R^2^ = 0.736 and 0.570 °brix) for RGB colour space (Table 5). Apart from the two-colour spaces, the MLR model also showed better performance for the HSL colour space in predicting TSS. The MLR model based on HSV colour space could predict TSS for training and testing stages with a 6.90% and 3.67% increase in R^2^ and a reduction of 12.97% and 5.26% in RMSE, respectively, compared to RGB based MLR model. Several studies predicted TSS of strawberries using different types of non-destructive techniques, with variable results [77,78,79]. Agulheiro-Santos et al. [79] established partial least squares regression (PLSR) based on near-infrared spectroscopy (NIRS) for TSS prediction of strawberries with determination of coefficients 0.669 and 0.520 in training and testing, respectively. A similar experiment was conducted by Shen et al. [77] and obtained a slightly better result in strawberry TSS prediction using partial least square discriminant analysis, with determination of coefficients 0.733 in testing. Though the performances of Vis-NIR spectroscopy were acceptable for the quantification of TSS, the main disadvantages of the technique are a lack of spatial information as well as expensive equipment required [80]. The observed and predicted datasets of TSS in RGB, HSV and HSL colour spaces for MLR model over the testing span were presented in scatter plots and time series graphs (Figure 6). Figure 6 illustrated that the observed and predicted values were fitted well and those were very close to the 1:1 line for HSV dataset compared to RGB and HSL, indicating MLR model had high accuracy in the estimation of TSS for HSV colour space. 

In terms of pH prediction, the highest R^2^ was found for HSV colour space in MLR model during the training (R^2^ = 0.742) and testing (R^2^ = 0.684), demonstrating that the MLR model can describe a maximum of 74.2% and 68.4% of the variations in the measured and predicted data in training and testing period, respectively. Among the three colour spaces, RGB colour space in MLR model obtained the lowest R^2^ and the highest RMSE in training (R^2^ = 0.705 and RMSE = 0.188) and testing (R^2^ = 0.643 and RMSE = 0.229) in predicting pH (Table 6). Aside from the two-colour spaces, the MLR model also performed better in predicting pH for the HSL colour space. When compared to the RGB-based MLR model, the HSV-based MLR model could predict pH in the training and testing stages with a 5.25% and 6.38% increase in R^2^ and a reduction of 3.20% and 6.11% in RMSE, respectively. Based on the outcomes of the model, it can be concluded that MLR model can predict pH using the colour features of strawberries. Guo et al. [78] related the colour feature to strawberry pH and showed that strawberries with different colour features represent the different levels of pH, demonstrating that colour features can provide supplementary, diverse and potentially effective information for determining pH in strawberries. Similar findings were also observed in fruits from other strawberry cultivars [6,17]. Figure 7 shows that the observed data were fitted well with the predicted data and the values were very close to the 1:1 line, representing the MLR model with HSV obtained better accuracy compared to RGB and HSL colour spaces in predicting pH.

### 3.5. SVM-R Model Performance for TSS and pH Prediction

The support vector machine regression (SVM-R) can fit well for non-linear data and results in better calibration and prediction with high R^2^ and low RMSE [23]. Since MLR is a linear based regression model, a non-linear regression model, i.e., SVM-R model, was carried out for prediction of TSS and pH of strawberries in the present study. The SVM-R model provided high accuracy for its non-linear approach, and several studies have used this technique to predict TSS and pH for a number of fruits [23,81]. Like MLR, SVM-R was also applied to the same data to compare the TSS and pH predictability of the three colour spaces. 

Based on the two statistical parameters, i.e., R^2^ and RMSE, the result of the current study showed that the SVM-R with HSV colour space provided better performance compared to SVM-R based on RGB and HSL colour space. Measured and predicted parameters of SVM-R models including R^2^ and RMSE for each colour space for TSS prediction are presented in Table 5. According to the SVM-R outcomes, the highest R^2^ (training = 0.841 and testing = 0.792) and the lowest RMSE (training = 0.437 and testing = 0.506) values were found for HSV colour space. This finding stated that the SVM-R model with HSV could explain a maximum 84.1% in training and 79.2% in testing period which can effectively predict the TSS of strawberries. On the other hand, the worst performance was found for the RGB-based SVM-R model (Table 5). In addition, the HSV-based SVM-R model could predict TSS in training and testing stages with a 2.70% and 5.46% increase in R^2^ and a 9.18 % and 8.67% reduction in RMSE, respectively compared to the RGB-based SVM-R model. Similar results were reported by Hernanz et al. [82], that applied multivariate statistical methods with different colour spaces to single out the colour features to correlate them with the TSS content. Some previous studies were conducted to compare the performance of RGB, HSV and HSL colour segmentation methods in object detection [83,84]. Mohd Ali et al. [83] stated that HSV colour algorithm achieved a better detection accuracy in objects compared to RGB colour space. Moreover, the predicted and measured data of TSS obtained from SVM-R model with RGB, HSV and HSL colour spaces were presented in scatter plots and time series graphs (Figure 8). According to Figure 8, the observed and predicted data were well-fitted and the values were very close to the 1:1 line for HSV, demonstrating SVM-R model had higher accuracy in the prediction of TSS for HSV compared to RGB and HSL colour space.

As with TSS prediction, pH prediction also followed similar results for the input parameters. A reasonable fit is observed for pH, using the SVM-R model with HSV colour space, with R^2^ of 0.788 and 0.726, and RMSE of 0.165 and 0.201 in training and testing period, respectively. The prediction performance of the pH of the collected strawberries with RGB colour feature was slightly lower compared to HSV and HSL based SVM-R model. The comparison evaluation metrics showed that the HSV based SVM-R model can predict a 5.35% and 6.37% increase in R^2^ and a reduction of 5.71% and 6.07% in RMSE in training and testing stages, respectively compared to RGB based SVM-R model. Moreover, the SVM-R model can explain a maximum of 78.8% and 72.6% of the variations in measured and predicted data in training and testing stages, respectively.

This is in concurrence with the results reported by Weng et al. [85], showing the determination coefficients: R^2^ = 0.744 in training and R^2^ = 0.660 in testing when the SVM with colour and textural features of strawberries were applied in pH prediction. Moreover, multivariate methods with spectra data, colour, texture and morphology of analytes extracted from the hyperspectral images were widely applied for pH determination [85]. Hu et al. [86] developed an advanced level hyperspectral imaging system for predicting internal properties of strawberries such as TSS and pH using a random frog based algorithm. Though these techniques sometimes achieved high performance in the prediction of fruit qualities, the main disadvantages of those non-destructive methods are highly expensive equipment needs to be used and the procedures are very complex which is sometimes inconceivable for farmhands for fruits quality detection and assessment. In the current study, we used SVM-R model in predicting pH where the observed and predicted pH data were fitted well and the values were very close to the 1:1 line for HSV colour space, suggesting the model had a high accuracy value (Figure 9). 

### 3.6. Comparison between MLR and SVM-R Model’s Performance with Colour Spaces 

When comparing all the developed machine learning models based on MLR and SVM-R algorithms with different colour spaces (RGB, HSV and HSL), the HSV colour space was the best for both MLR and SVM-R models for the prediction of TSS and pH values. However, we also observed that the HSL colour space also achieved a relatively good prediction accuracy for TSS and pH values. This finding indicated that peel colour changes in Seolhyang strawberries had a certain relationship with the internal quality parameters of fruit such as pH and TSS. Nunes et al. [11] demonstrated that the colour development of ‘Oso Grande’ strawberries increased during the ripening which had a direct relationship to internal quality parameters because all quality parameters are highly influenced by the ripening time. Similar changes were also observed in ‘Sweet Charlie’ strawberry cultivar [87]. Moreover, in the image processing technique, it is a critical issue to select a suitable colour space from a large variety of available colour spaces that can produce the best prediction result for models. With a comparison of the prediction among RGB, HSV and HSL colour spaces, we found that HSV was better than RGB and HSL regardless of which model was used. One of the main reasons behind this may be three channels, i.e., Hue, Saturation and Value, in HSV are approximately orthogonal in shape that can reduce more the influence of other channel changes and avoid the channel overlapping compared to RGB and HSL colour spaces [88]. This may be one of the main reasons why most of the studies used HSV colour space for internal quality assessment in fruit and object detection methods [83,84,88].

In addition, SVM-R models had a little better performance than the MLR in TSS and pH prediction (Figure 10). Based on the statistical evaluation parameters (R^2^ and RMSE), the findings of the study showed that the selected SVM-R model based on HSV colour space could predict TSS with 2.43% and 3.80% increase in R^2^ and a reduction of 5.62% and 6.30% in RMSE in training and testing stages, respectively compared to HSV based MLR model. In terms of pH prediction, SVM-R model with HSV could predict with 6.20% and 6.14% increase in R^2^ and a reduction of 9.34% and 6.51% in RMSE in training and testing stages, respectively, compared to MLR model with HSV colour space. This reason might be a certain portion of the relationship between input and predicted parameters existed in a non-linear relationship and the non-linear relation could contribute to the better performance of the SVM-R model. Fernández-Ahumada et al. [85] noted that the higher ability of the SVM-R model compared with the MLR model due to capturing a strong nonlinearity and a large number of data. Moreover, results obtained from the cumulative distribution function revealed that the SVM-R model obtained a residual value of 77.5% of the actual and predicted data of TSS between the ranges of −0.5 to 0.5, whereas it was 61.67% for the MLR model in the same range (Figure 11b). Like TSS, the SVM-R model has a residual of 76.67% of the actual and predicted data for pH between the ranges of −0.2 to 0.2 whereas, it was 60.83% for MLR model. In addition, Figure 11 illustrated a non-linear relationship between the predicted (TSS and pH) and measured variables (channels in RGB, HSV and HSL colour space).

### 3.7. Sensitivity Analysis of Colour Spaces 

To find out the effect of each channel, i.e., Hue, Saturation and Value, of HSV colour spaces, a sensitivity test was carried out for the SVM-R and MLR models. The ability of SVM-R and MLR models little decreased when they were run without the S value during the pH prediction (Figure 12). As shown in Figure 12, without the S as an input value, SVM-R and MLR obtained the lowest R^2^ (0.717 and 0.661) and the highest RMSE (0.196 and 0.223) in the testing stage, respectively. Therefore, it can be said that S was the most influential input variable in predicting pH, followed by V and H, respectively. Like pH, sensitivity test was conducted in TSS prediction using the SVM-R and MLR model with HSV colour space. The findings of the sensitivity also showed that excluding S, the performance of both models decreased, indicating S was the most important channel in HSV colour space in predicting TSS. According to the findings of the sensitivity analysis, it can be concluded that the S and H were the most and least influential factors, corresponding to predict TSS and pH for both models. Some published articles also reported similar results on the importance of colour channels in object detection [89,90].

## 4. Conclusions

The study was carried out to evaluate the application of machine learning models for the TSS and pH prediction of strawberries using image processing technique. The results showed that biometrical characteristics, i.e., length, diameters, weight of strawberries, TSS and pH, and channels of each colour space changed significantly at different levels from the white to dark red stage. The channels of RGB, HSV and HSL colour spaces were used as input variables for TSS and pH prediction models and the performance of both models were evaluated using two statistical parameters (R^2^ and RMSE). The results indicated that all SVM-R models performed slightly better than MLR models for the prediction of TSS and pH values, demonstrating a non-linear relationship between the explanatory and response variables. Simultaneously, the results also showed that the HSV was the best one among the three colour spaces for the developed MLR and SVM-R models. The HSV based SVM-R model could explain a maximum of 84.1% and 79.2% for TSS and 78.8% and 72.6% for pH of the variations in measured and predicted data in the training and testing stages, respectively. Moreover, sensitivity analysis showed that the S (Saturation) and H (Hue) were the most and least influential channels in HSV colour space to predict TSS and pH values. In future, more samples with different strawberry cultivars may need to be studied to further measure whether the SVM-R models performed better than other regression-based models, such as Random Forest, for the prediction of more internal qualities of fruits. The present study suggested that this new approach is a promising non-destructive, time-efficient and less expensive equipment required for rapid monitoring; therefore, our next research will focus on its application in the real field.

## Figures and Tables

**Figure 1 foods-11-02086-f001:**
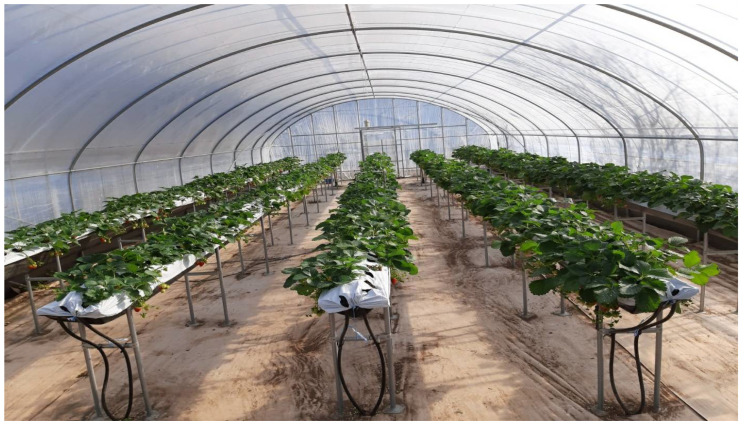
Experimental greenhouse for strawberry cultivation.

**Figure 2 foods-11-02086-f002:**
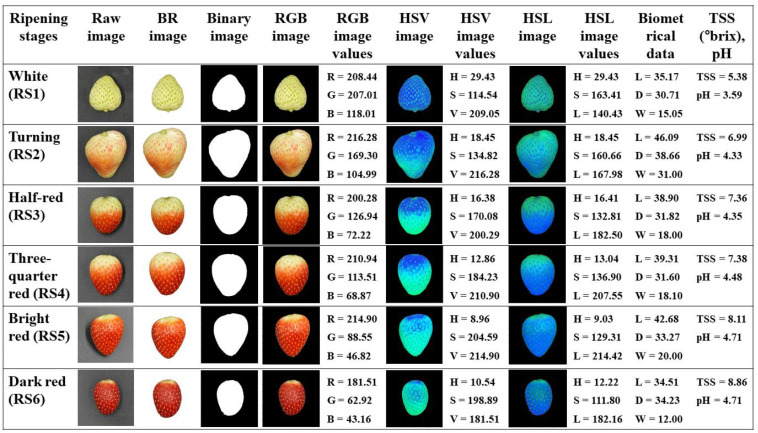
Images of strawberries with background removal (BR), binary, RGB, HSV and HSL colour spaces in six ripening stages. Biometrical data (Length (L in mm), diameter (D in mm) and weight (W in g)) and TSS (°brix) with pH are shown in the two right columns, respectively.

**Figure 3 foods-11-02086-f003:**
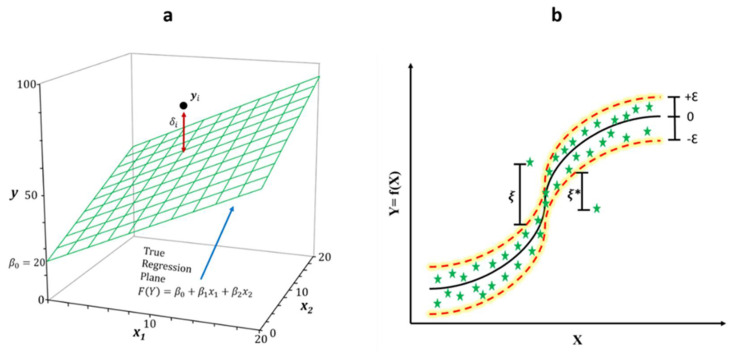
Diagrams show the structure of (**a**) multiple linear regression (MLR) and (**b**) support vector machine regression (SVM-R).

**Figure 4 foods-11-02086-f004:**
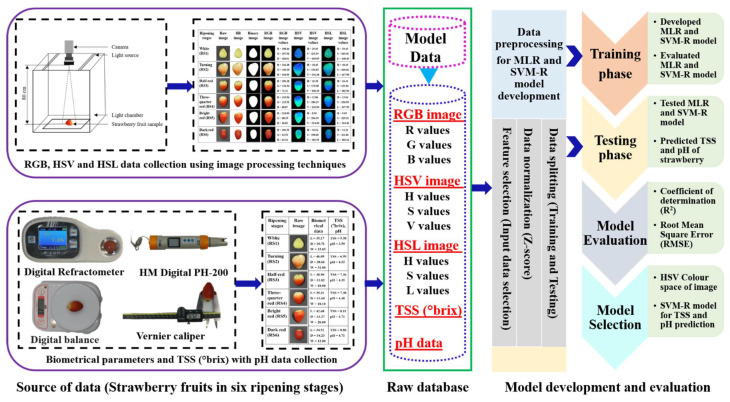
Flow diagram of prediction of total soluble solids (TSS) and pH content of strawberries in six ripening stages with RGB, HSV and HSL colour spaces using multiple linear regression (MLR) and support vector machine regression (SVM-R).

**Figure 5 foods-11-02086-f005:**
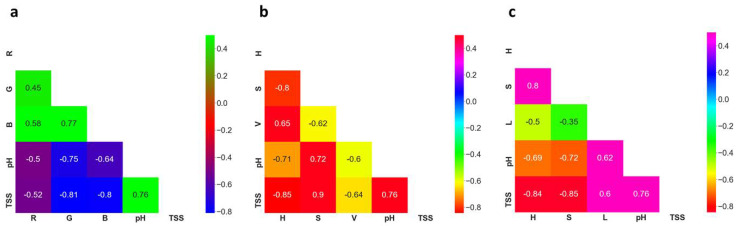
Pearson correlation coefficients for the relationship between predictor and response variables were used in this study. Heat map (**a**) represents the relationship between colour space values of RGB images and response variables (TSS and pH); (**b**) represents the relationship between colour space values of HSV images and response variables (TSS and pH) and (**c**) represents the relationship between colour space values of HSL images and response variables (TSS and pH). The colour intensity in the matrix is proportional to the correlation coefficients.

**Figure 6 foods-11-02086-f006:**
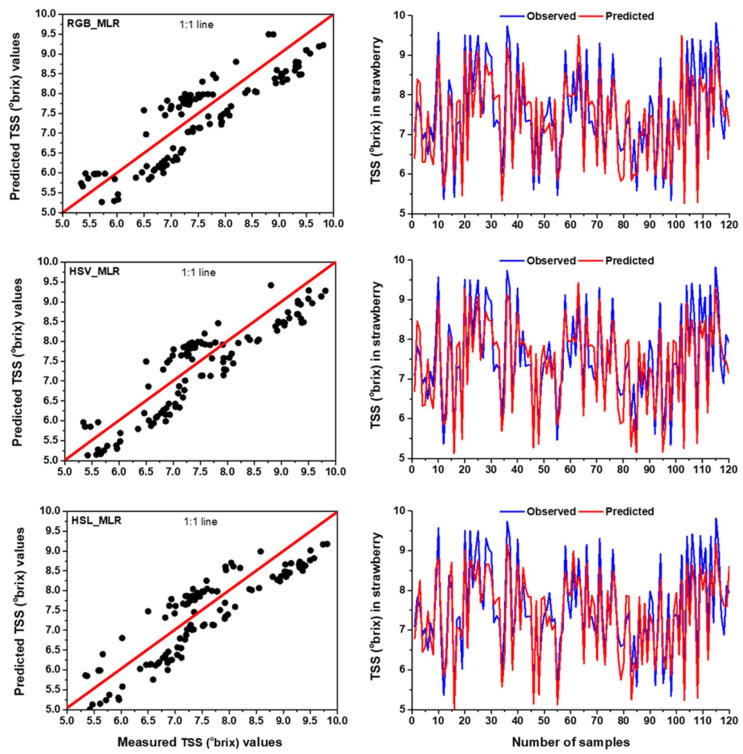
The comparison results between measured and predicted values by MLR model with RGB, HSV and HSL colour spaces for TSS prediction in the testing period using time series and scatter plots with 1:1 line.

**Figure 7 foods-11-02086-f007:**
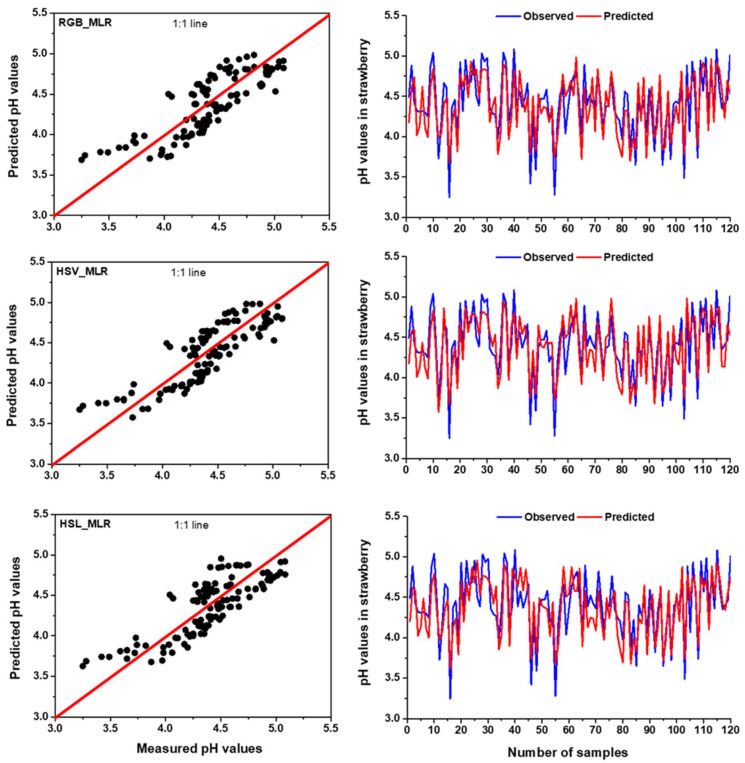
The comparison results between measured and predicted values by MLR model with RGB, HSV and HSL colour spaces for pH prediction in the testing period using time series and scatter plots with 1:1 line.

**Figure 8 foods-11-02086-f008:**
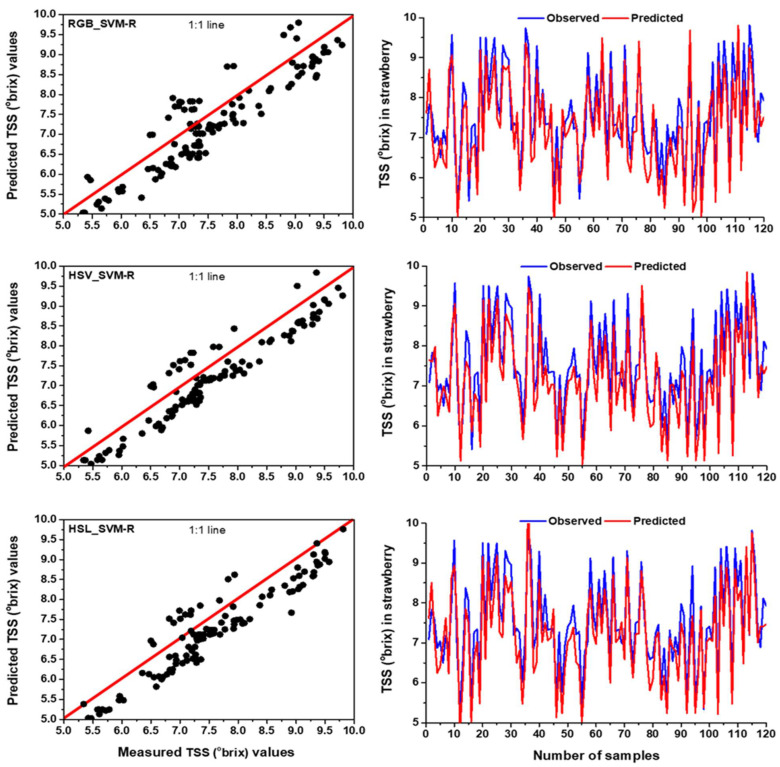
The comparison results between measured and predicted values by SVM-R model with RGB, HSV and HSL colour spaces for TSS prediction in the testing period using time series and scatter plots with 1:1 line.

**Figure 9 foods-11-02086-f009:**
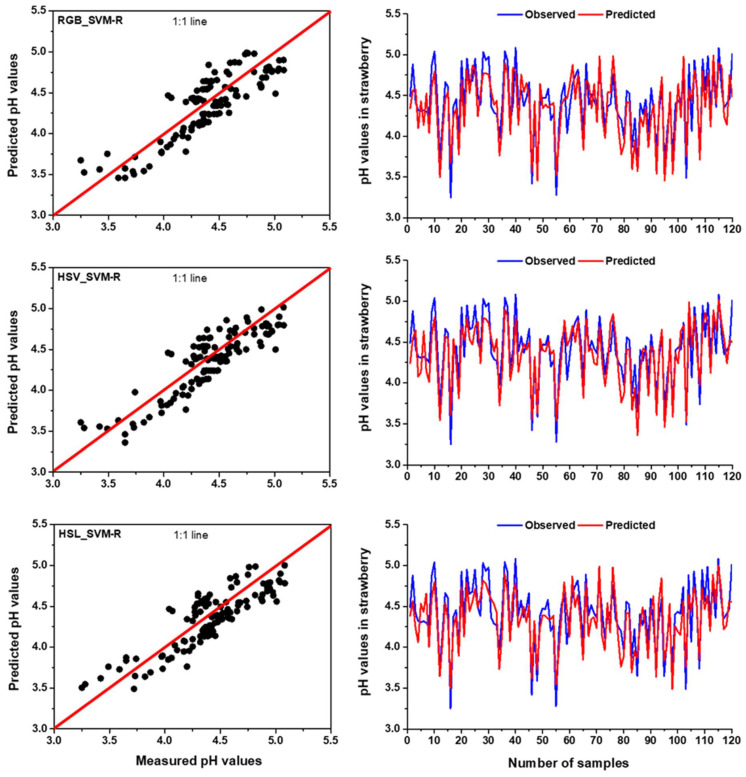
The comparison results between measured and predicted values by SVM-R model with RGB, HSV and HSL colour spaces for pH prediction in the testing period using time series and scatter plots with 1:1 line.

**Figure 10 foods-11-02086-f010:**
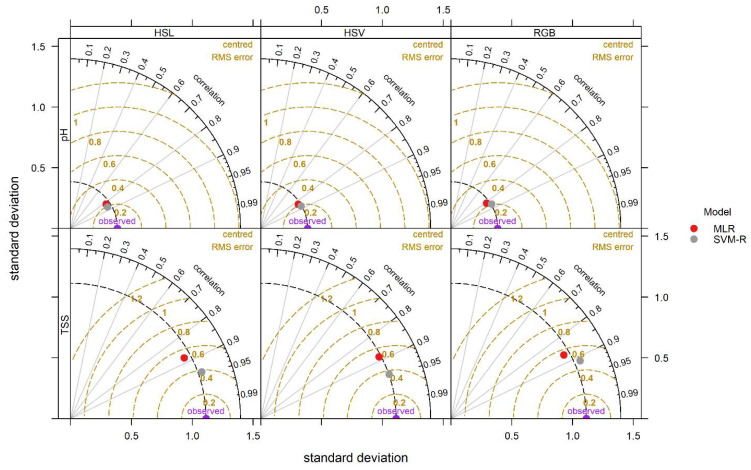
Taylor diagram of testing results of MLR and SVM-R models with RGB, HSV and HSL datasets for TSS and pH prediction.

**Figure 11 foods-11-02086-f011:**
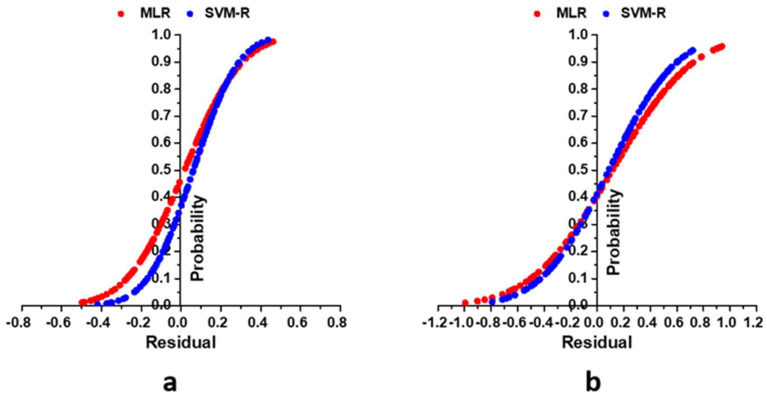
Cumulative distribution analysis using MLR and SVM-R models, (**a**) graph calculated from measured and predicted pH values (**b**) graph calculated from measured and predicted TSS content data during the testing period.

**Figure 12 foods-11-02086-f012:**
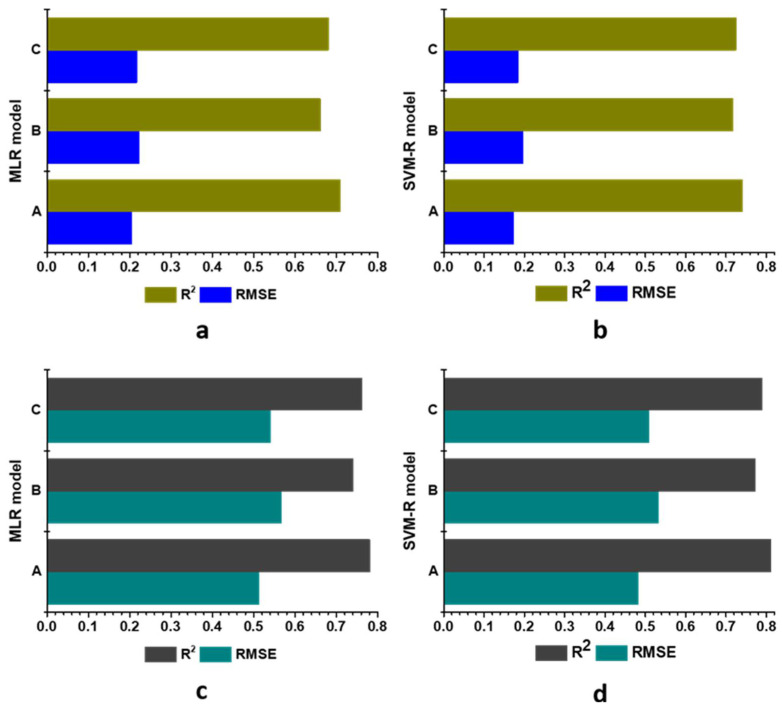
Sensitivity analysis of the HSV colour spaces on pH and TSS of strawberry fruits using MLR and SVM-R models. A: MLR/SVM-R model without V values; B: MLR/SVM-R model without S values; C: MLR/SVM-R model without H values. Sensitivity analysis of the HSV colour spaces on pH using MLR model (**a**); Sensitivity analysis of the HSV colour spaces on pH using SVM-R model (**b**); Sensitivity analysis of the HSV colour spaces on TSS using MLR model (**c**); Sensitivity analysis of the HSV colour spaces on TSS using SVM-R model (**d**).

**Table 1 foods-11-02086-t001:** Biometrical characteristics, total soluble solids (TSS) and pH in strawberry fruits at six stages of ripening.

Ripening Stage	Length (mm)	Diameter (mm)	Weight (g)	TSS (°brix)	pH
White (RS1)	36.72 ± 2.12 ^a^	30.79 ± 1.42 ^a^	15.33 ± 1.84 ^a^	5.99 ± 0.48 ^a^	3.85 ± 0.24 ^a^
Turning (RS2)	37.34 ± 3.36 ^a^	31.57 ± 2.47 ^a^	17.17 ± 4.06 ^b^	6.69 ± 0.36 ^b^	4.17 ± 0.18 ^b^
Half-red (RS3)	39.35 ± 2.61 ^b^	31.66 ± 2.79 ^ab^	18.31 ± 3.43 ^b^	7.18 ± 0.34 ^c^	4.34 ± 0.06 ^c^
Three-quarter red (RS4)	41.01 ± 4.17 ^c^	31.11 ± 1.71 ^a^	18.34 ± 2.57 ^b^	7.29 ± 0.22 ^c^	4.46 ± 0.12 ^d^
Bright red (RS5)	43.30 ± 2.92 ^d^	32.48 ± 1.73 ^bc^	19.97 ± 2.54 ^c^	8.24 ± 0.41 ^d^	4.59 ± 0.21 ^e^
Dark red (RS6)	42.71 ± 4.03 ^d^	33.00 ± 2.70 ^c^	20.58 ± 4.76 ^c^	9.19 ± 0.35 ^e^	4.87 ± 0.16 ^f^
Overall in six stages	40.07 ± 4.12	31.77 ± 2.32	18.28 ± 3.76	7.43 ± 1.10	4.38 ± 0.36

Values denote the mean of 100 individual measurements of strawberries (*n* = 100) in each ripening stage ± the standard deviation. Values not mentioned by the same letter in the same column are significantly different (*p* < 0.05).

**Table 2 foods-11-02086-t002:** Values of RGB (Red: *R*, Green: *G*, Blue: *B*) colour space in strawberry fruits at six stages of ripening.

Ripening Stage	RGB Colour Space		
	*R*	*G*	*B*
RS1	203.54 ± 6.52 ^a^	200.66 ± 6.52 ^a^	120.02 ± 11.18 ^a^
RS2	208.84 ± 4.18 ^b^	183.49 ± 11.70 ^b^	111.29 ± 10.99 ^b^
RS3	207.42 ± 9.86 ^b^	133.60 ± 9.02 ^c^	78.17 ± 10.32 ^c^
RS4	208.06 ± 8.77 ^b^	118.09 ± 7.02 ^d^	68.75 ± 5.21 ^d^
RS5	209.14 ± 5.50 ^b^	89.09 ± 8.07 ^e^	46.86 ± 5.91 ^e^
RS6	181.80 ± 6.34 ^c^	59.21 ± 4.98 ^f^	39.57 ± 3.64 ^f^
Overall	203.13 ± 12.04	130.69 ± 50.18	77.44 ± 31.19

Values of RGB colour spaces collected from 100 individual collections of strawberry fruit (*n* = 100) images in each ripening stage denote the mean ± the standard deviation. Values not mentioned by the same letter in the same column are significantly different (*p* < 0.05).

**Table 3 foods-11-02086-t003:** Values of HSV (Hue: *H*, Saturation: *S*, Value: *V*) colour space in strawberry fruits at six stages of ripening.

Ripening Stage	HSV Colour Space		
	*H*	*S*	*V*
RS1	28.87 ± 0.26 ^a^	110.29 ± 10.45 ^a^	203.88 ± 6.52 ^a^
RS2	24.17 ± 2.14 ^b^	125.88 ± 13.52 ^b^	208.92 ± 4.17 ^b^
RS3	16.20 ± 1.03 ^c^	171.47 ± 10.70 ^c^	207.43 ± 9.85 ^b^
RS4	13.85 ± 0.85 ^d^	183.39 ± 4.58 ^d^	208.08 ± 8.77 ^b^
RS5	9.76 ± 0.87 ^e^	206.12 ± 5.89 ^e^	209.14 ± 5.51 ^b^
RS6	9.98 ± 1.54 ^f^	207.08 ± 5.18 ^e^	181.80 ± 6.34 ^c^
Overall	16.95 ± 7.42	167.37 ± 38.36	203.21 ± 12.05

Values of HSV colour spaces collected from 100 individual collections of strawberry fruit (*n* = 100) images in each ripening stage denote the mean ± the standard deviation. Values not mentioned by the same letter in the same column are significantly different (*p* < 0.05).

**Table 4 foods-11-02086-t004:** Values of HSL (Hue: *H*, Saturation: *S*, Lightness: *L*) colour space in strawberry fruits at six stages of ripening.

Ripening Stage	HSL Colour Space		
	*H*	*S*	*L*
RS1	28.87 ± 0.26 ^a^	161.65 ± 8.79 ^a^	133.34 ± 5.47 ^a^
RS2	24.17 ± 2.14 ^b^	159.48 ± 2.14 ^a^	153.85 ± 11.05 ^b^
RS3	16.24 ± 1.02 ^c^	139.83 ± 9.73 ^b^	190.73 ± 10.93 ^c^
RS4	13.85 ± 0.85 ^d^	135.28 ± 6.64 ^c^	201.48 ± 6.58 ^d^
RS5	9.92 ± 0.87 ^e^	125.63 ± 5.15 ^d^	212.64 ± 3.06 ^e^
RS6	11.03 ± 2.34 ^f^	109.21 ± 4.16 ^e^	195.71 ± 7.24 ^f^
Overall	17.35 ± 7.08	138.51 ± 19.68	181.29 ± 29.21

Values HSL of colour spaces collected from 100 individual collections of strawberry fruit (*n* = 100) images in each ripening stage denote the mean ± the standard deviation. Values not mentioned by the same letter in the same column are significantly different (*p* < 0.05).

**Table 5 foods-11-02086-t005:** Performance metrics (R^2^ and RMSE) of the models for predicting TSS of strawberry fruits during the testing and training period. The values in italics indicate the best results in each model among the three colour spaces.

Model Name	Dataset	Training		Testing	
		R^2^	RMSE	R^2^	RMSE
MLR	RGB	0.768	0.532	0.736	0.570
	HSV	*0.821*	*0.463*	*0.763*	*0.540*
	HSL	0.808	0.479	0.756	0.547
SVM-R	RGB	0.819	0.481	0.751	0.554
	HSV	*0.841*	*0.437*	*0.792*	*0.506*
	HSL	0.833	0.447	0.784	0.515

**Table 6 foods-11-02086-t006:** Performance metrics (R^2^ and RMSE) of the models for predicting the pH of strawberry fruits during the testing and training period. The values in italics indicate the best results in each model among the three colour spaces.

Model Name	Dataset	Training		Testing	
		R^2^	RMSE	R^2^	RMSE
MLR	RGB	0.705	0.188	0.643	0.229
	HSV	*0.742*	*0.182*	*0.684*	*0.215*
	HSL	0.722	0.189	0.655	0.225
SVM-R	RGB	0.748	0.175	0.688	0.214
	HSV	*0.788*	*0.165*	*0.726*	*0.201*
	HSL	0.774	0.171	0.714	0.204

## Data Availability

The datasets generated during and/or analysed during the current study are available from the corresponding author on reasonable request.

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
