# Peer review of "Prediction of Total Soluble Solids and pH of Strawberry Fruits Using RGB, HSV and HSL Colour Spaces and Machine Learning Models"

_foods, 2022, doi:10.3390/foods11142086_

Round 1

Reviewer 1 Report

The manuscript is written with clear understanding of the project addressed. However, there are major concerns that need to be addressed to enhance the quality of the manuscript. My specific comments are as follows:

Introduction:

L40-43: “Hence, the main aim of this article was to develop machine learning-based models using Red, Blue, and Green (RGB), Hue, Saturation, and Value (HSV) and Hue, Saturation, and Lightness (HSL) colour space’s values of strawberry images..” Put at the end of last paragraph

L90: “…several image process techniques have been developed and applied successfully for…” Discuss several image processing techniques used

Based on your objectives, please compare how your study is different from those that have already been published

Materials and methods:

L140: The whole input sample image was used as the region of interest (ROI) to extract color and texture features.” Explain the process for ROI extraction

Is there any image preprocessing conducted for each image?

Specify the split ratio training and testing datasets.

Add data analysis part.

Results and discussion:

L304: “TSS in strawberries increased continuously from the white to dark red ripening stage with statistically significant.” State the range of TSS

State the range of pH values

Which one produce the best results?

Instead of mentioning the results, the authors should justify/explain the findings

Conclusions:

L589: “ In this study, 600 strawberries were collected in six ripening stages from an experimental greenhouse to measure the biometrical characteristics, TSS and pH as well as to develop MLR and SVM-R based regression models using colour space values of strawberry’s images.” Delete

Justify the main finding of your study.

Add recommendation for future studies.

General comments:

Please check the reference styles and grammar of the manuscript.

Author Response

Dear reviewer

Thank you for considering our manuscript and for the precious time you spent reviewing it. We made the changes in the manuscript according to the suggestion and comments provided. Overall, we corrected and tried to clarify the mistakes identified throughout the manuscript.

Reviewer 2 Report

Prediction of Total Soluble Solids and pH of Strawberry Fruits using RGB, HSV and HSL Colour Spaces and Machine Learning Modelsis very interesting and well written

 Presented manuscript is on high scientific level. The manuscript authors the introduction section presented the current state of knowledge on the experimental design. The topic is a new, as well present a very important aspects of quality of strawberry and development of non-destructive methods for is assessment.

 The summary. Authors give a short presentation of manuscript. This part of manuscript is well constructed.  

 The Introduction section includes all necessary information about examined objects and problems.

 Page 1, line 36: …is: and vitamin [4]. What kind of vitamin Authors mind? Probably it should be vitamins (plural) or one of the most important vitamin C. - please correct it properly.

 Material and method section is well constructed and contain all necessary information.

 Results

 All figures and tables are clear and properly presented

 The discussion section presents a good comparison of the obtained results. Presented conclusions are corresponding with all information presented via Authors’ in manuscript text.

 General opinion:  After carefully manuscript reading, I think, that presented experiment is a valuable. In my opinion Manuscript should be corrected according to my pointing and remarks.

Author Response

Dear reviewer,

Thank you for considering our manuscript and for the precious time you spent reviewing it. We made the changes in the manuscript according to the suggestion and comments provided. Overall, we corrected and tried to clarify the mistakes identified throughout the manuscript.

Reviewer 3 Report

Minor revision. Some suggestions below:

A more attractive and concise title should be used.

In introduction, some lines on description of main nutrients and bioactive components of strawberry should be added.

Lines 34-78 should be better summarized.

Lines 79-121 should be summarized.

The advantages of use of NIR should be better marked.

The description of input datasets for model development should be implemented.

The comparison between the two models should be better described in the text.

The results of sensitivity analysis of colour spaces should be better described.

Author Response

(The authors gave the same response as above.)

Reviewer 4 Report

This study aimed to determine TSS and pH of strawberries using a RGB camera and SVM and MLR models.

1- Several papers have been published discussing similar topics. This paper does not possess any particular advantages over the following studies:

Mohammadi, V., Kheiralipour, K., & Ghasemi-Varnamkhasti, M. (2015). Detecting maturity of persimmon fruit based on image processing technique. Scientia Horticulturae, 184, 123-128.

Saad, A. M., Ibrahim, A., & El-Bialee, N. (2016). Internal quality assessment of tomato fruits using image color analysis. Agricultural Engineering International: CIGR Journal, 18(1), 339-352.

Cho, B. H., & Koseki, S. (2021). Determination of banana quality indices during the ripening process at different temperatures using smartphone images and an artificial neural network. Scientia Horticulturae, 288, 110382.

Villaseñor-Aguilar, M. J., Bravo-Sánchez, M. G., Padilla-Medina, J. A., Vázquez-Vera, J. L., Guevara-González, R. G., García-Rodríguez, F. J., & Barranco-Gutiérrez, A. I. (2020). A maturity estimation of bell pepper (Capsicum annuum L.) by artificial vision system for quality control. Applied Sciences, 10(15), 5097.

2- The development of a smartphone application for detecting quality indices in natural light conditions would be an interesting research project. However, the authors did not address this issue.

Author Response

(The authors gave the same response as above.)

Round 2

Reviewer 4 Report

The article is well organized and presented, and the authors discuss their findings in a clear and concise manner. However, I still believe that this study does not provide any novelty or advantage over other similar studies.

I can only recommend that the paper be decided based on the editorial assessment.